

# Insights into the nutritional prevention of macular degeneration based on a comparative topic modeling approach

Lucas Jacaruso

University of Southern California, Los Angeles, CA, United States of America

## ABSTRACT

Topic modeling and text mining are subsets of natural language processing (NLP) with relevance for conducting meta-analysis (MA) and systematic review (SR). For evidence synthesis, the above NLP methods are conventionally used for topic-specific literature searches or extracting values from reports to automate essential phases of SR and MA. Instead, this work proposes a comparative topic modeling approach to analyze reports of contradictory results on the same general research question. Specifically, the objective is to identify topics exhibiting distinct associations with significant results for an outcome of interest by ranking them according to their proportional occurrence in (and consistency of distribution across) reports of significant effects. Macular degeneration (MD) is a disease that affects millions of people annually, causing vision loss. Augmenting evidence synthesis to provide insight into MD prevention is therefore of central interest in this article. The proposed method was tested on broad-scope studies addressing whether supplemental nutritional compounds significantly benefit macular degeneration. Six compounds were identified as having a particular association with reports of significant results for benefiting MD. Four of these were further supported in terms of effectiveness upon conducting a follow-up literature search for validation (omega-3 fatty acids, copper, zeaxanthin, and nitrates). The two not supported by the follow-up literature search (niacin and molybdenum) also had scores in the lowest range under the proposed scoring system. Results therefore suggest that the proposed method's score for a given topic may be a viable proxy for its degree of association with the outcome of interest, and can be helpful in the systematic search for potentially causal relationships. Further, the compounds identified by the proposed method were not simultaneously captured as salient topics by state-of-the-art topic models that leverage document and word embeddings (Top2Vec) and transformer models (BERTopic). These results underpin the proposed method's potential to add specificity in understanding effects from broad-scope reports, elucidate topics of interest for future research, and guide evidence synthesis in a scalable way. All of this is accomplished while yielding valuable and actionable insights into the prevention of MD.

Corresponding author
Lucas Jacaruso,
jacaruso33@gmail.com

## INTRODUCTION

### Background and significance

Very rarely in science is consensus derived from a single report. Making sense of the "evidence landscape" on any research question often requires synthesizing results across multiple studies to elucidate the prevailing direction of known facts. Two main types of evidence synthesis are standard in science. One is meta-analysis (MA), the statistical combination of results across multiple studies. The second is systematic review (SR), the collation of relevant evidence from a sample of studies (without performing a statistical unification). Natural language processing (*Chowdhary, 2020*) (NLP) is the field dealing with the computational understanding (and, more recently, generation) of text and speech. NLP can support MA and SR by automating literature searches and fact/figure extraction from published studies.

Report aggregation isn't only valuable in the context of peer-reviewed studies. Unstructured anecdotal reports have also been useful in validating research directions across domains, for instance, regarding the potential therapeutic properties of cannabinoids for central nervous system disorders (*Croxford, 2003*) and the role of apex predators in sensitive ecosystems (*Somaweera et al., 2020*). In some cases, NLP has proven helpful in analyzing themes in anecdotal reports, such as mental health outcomes during pandemics and extracting insight from patient reports (*Marshall et al., 2022*; *Van Buchem et al., 2022*). In this light, collating reports is sometimes inherent to the discovery phase of new phenomena worth studying, with anecdotal observations serving as directional indicators for further research. Indeed, some domains can only be explored *via* sophisticated instrumentation and the application of advanced theory. In other domains, however, patient experiences or eyewitness testimony can have a real bearing on unanswered questions. More examples include tracking the benefits/side effects of new treatments, understanding symptoms of emerging pathogen variants, observing the effects of climate change on local ecosystems, studying unidentified phenomena in nature, comprehending the psychological impacts of new technologies, and more.

### Relevant literature

Within the scope of this study, the focus will be on published literature rather than anecdotal reports (although the proposed method can theoretically have applications for both). In the context of MA and SR, the application of NLP has arguably fallen into two main discernible subcategories: systematic data extraction and literature search (though not limited to these). Text mining is a subset of NLP dealing with the computational extraction of raw data from text (*Hassani et al., 2020*), *e.g.*, numerical values, figures, and names. Various examples exist in the literature of capturing information from papers *via* text mining for performing MA. *Mutinda et al. (2022)* propose a named entity recognition model to identify participants, intervention, control, and outcomes (PICO) from clinical trials. In another domain, the Neurosynth Corpus (*Neurosynth, 2023*) is created *via* text mining, namely by tying keyword mentions to fMRI images from a large sample of studies to assemble a library of neuroimaging representations associated with cognitive, physical, and emotional states. Additionally, text mining is leveraged to gather data for MA to

identify xerostomia drug targets (*Beckman et al., 2022*), evaluate the efficacy of educational software (*Costa et al., 2020*), and augment biomarker discovery (*Bhatnagar et al., 2022*).

The other preeminent category is NLP-enabled methods for literature search. *Feng, Chiam & Lo (2017)* explore the value of NLP for automating parts of the SR process. They find that the study selection phase of SR most commonly benefits from text-mining approaches compared to later stages of SR, such as assessing study quality or other types of screening. *Marshall et al. (2018)* introduce a freely available framework for searching studies while filtering for randomized controlled trials (RCTs). They achieve this by using the support vector machine (SVM) (*Pisner & Schnyer, 2020*) and convolutional neural network (CNN) (*Albawi, Mohammed & Al-Zawi, 2017*) machine learning architectures for document classification as RCT *vs.* non-RCT. *Cohen et al. (2015)* introduce a system that uses the Liblinear (*Fan et al., 2008*) fast linear implementation of the SVM to identify RCTs. *O'Mara-Eves et al. (2015)* show how NLP can be used to prioritize the order of study retrieval based on the likelihood of topic relevance. In another recent study, *Kim & Gil (2019)* demonstrate a topic-modeling implementation for study retrieval involving the Term-Frequency Inverse-Document-Frequency algorithm (TF-IDF). Topic modeling can be considered the subset of text mining that deals with identifying the overarching subject matter of documents, with obvious value for literature search and relevant information retrieval. Based on a review of related work, it is clear that NLP and topic modeling offer practical value for key phases of MA and SR.

Various approaches for topic modeling have been widely adopted, including latent Dirichlet allocation (LDA) (*Blei, Ng & Jordan, 2003*), latent semantic analysis (LSA) (*Dumais, 2004*), and TF-IDF. Latent Dirichlet allocation is frequently used for supporting MA and SR (*Mo, Kontonatsios & Ananiadou, 2015*; *Choi, Lee & Sohn, 2017*; *Asmussen & Møller, 2019*). TF-IDF proves to be an indispensable measure that is still applied (either with or without modifications) in a wide range of novel use cases in recently proposed papers (*Kim & Gil, 2019*; *Gomes, da Silva Torres & Côrtes, 2023*; *Jalilifard et al., 2021*; *Rawat et al., 2021*; *Yuan et al., 2020*). TF-IDF is therefore still a current topic in the NLP field as indicated by the relative recency of the literature leveraging this algorithm. In NLP terminology, "documents" refers to individual entries (*e.g.*, papers, social media posts, or patient reports), whereas the "corpus" is the broader dataset that is the collection of documents. TF-IDF captures the salient topics of a given document by identifying the most ubiquitous terms within the document that are simultaneously rare throughout the larger corpus. The highest-ranking terms are most likely representative of the document's subject matter. Intuitively, if "chromosome" appears frequently in a specific document but not in the rest of the corpus, it will have a high score. Conversely, the word "and" will have a low score since it occurs frequently in both the document and across the corpus and does not represent a meaningful topic. While topic modeling methods are abundant in the literature (including for supporting SR and MA, as demonstrated earlier), this article explores a nuanced type of topic modeling not well-explored in the literature, the goal being to yield insights into the prevention of debilitating diseases.

## Contributions

NLP methods for MA or SR typically focus on literature search (*via* topic modeling and classification) or text mining for facts and figures. Instead, the interest of this article is a topic modeling approach for finding differentiating subtopics in reports containing contradictory findings to others on the same question, presenting unique opportunities to yield insight and advance understanding. Seemingly conflicting results can be found on a vast number of questions across medical literature (*Lamers et al., 2021*), for instance, whether exposure therapy is effective for PTSD (*Huang et al., 2022*; *Markowitz & Fanselow, 2020*), the role of meat consumption in a healthy diet (*Rohrmann et al., 2013*; *Leroy & Cofnas, 2019*) and many other examples. Indeed, disparities in outcome are often attributable to general methodology, study design, or demographics studied. However, in many cases, specific compounds, dosages, practices, or other discoverable factors may be associated with outcomes consistently enough to warrant further study. Consider resveratrol, a compound studied for various health benefits (including antioxidant and anti-tumor activity). Studies with the most significant results tended to administer resveratrol with liposomes since it is a fat-soluble compound (with limited water solubility), as found in the SR conducted by *Amri et al. (2012)*. This detail (*i.e.,* administration with liposomes) may be responsible for divergent outcomes in early studies examining the benefits of resveratrol before there was considerable literature volume on this topic. Another example is the question of whether nutritional supplementation prevents macular degeneration (MD) in any significant way. MD prevention is chosen as the area of focus since it affects millions of people annually, and its treatment and prevention represents an ongoing medical research avenue. Individual studies may include different nutritional compounds than others. If particular compounds are mentioned more frequently in the context of significant findings for the prevention of MD, those compounds would be of interest for further study as potential causal factors for the desired outcome. This article demonstrates an NLP-enabled approach for finding such factors. Going forward, these factors will simply be called "potential determinants."

While TF-IDF models topics by using term frequencies within a document and across a corpus (as explained above), a new approach is proposed to identify potential determinants. Unlike most current topic modeling techniques, the proposed approach assumes two corpora, one containing studies showing significant positive effects and another containing studies showing insignificant/negative effects for a relationship of interest (*e.g.,* nutritional compounds and the prevention of MD). These will be called the "positive" and "negative" corpus for simplicity. Finding potential determinants is not as trivial as finding topics unique to the positive corpus (since some overlap between corpora is expected). At the same time, it isn't as simple as comparing a topic's frequency between the corpora; the consistency of a topic's distribution across documents in the positive corpus also signals its interest as a potential determinant (to ensure a higher frequency is not simply attributable to a single document). Some terms may merely be tangentially mentioned in the report (rather than being directly studied), confounding the matter even more. Further, potential determinants may not necessarily be the most frequently discussed topics in the positive corpus while still having a meaningful association with the outcome of interest. Therefore, while methods exist in the literature for directly comparing topics

between datasets (*e.g.*, for news event comparison (*Hua et al., 2020*)), a specially conceived approach is needed to qualify topics as potential determinants in the context of divergent study results. The proposed method takes some basic intuitions from TF-IDF, generating a score for each term in the positive corpus. However, the proposed approach fundamentally differs from TF-IDF while utilizing the same attributes. For a given term, the proposed topic significance score is a function of its proportional occurrence in the positive corpus (out of all occurrences across both corpora) and its distribution across documents within the positive corpus. Candidate topics are also filtered for membership within a category of interest. The latter criterion ensures that results are focused within a particular class (*e.g.*, nutritional compounds), enabling a specific and meaningful search for potential determinants. While large language models may seem helpful for the same use case, they can produce spurious information in their current form and may have limited explainability (*Azamfirei, SR & Fackler, 2023*; *Krause et al., 2023*). An explainable method is of current interest (not to rule out the promise of LLMs for this use case in the future, perhaps in tandem with methods like the proposed approach).

The proposed method will be tested on papers addressing the *general question* of whether nutritional compounds can prevent the development or progression of MD in a significant way. In other words, it will be applied to research featuring breadth and variety in the compounds it tests/reviews (*e.g.*, multivitamins, diets, or supplement formulations) rather than narrow scope (*e.g.*, one particular compound). The objective of the proposed method is to extract specific potential determinants from the broad-scope reports. Validation will consist of a retrospective literature search on each identified compound. Suppose the results of the proposed method are supportable for benefiting MD (according to narrow-scope follow-up studies that address each compound); this would substantiate the proposed method's ability to pinpoint relevant topics for desired outcomes from a wide range of candidates, inform new research directions, and support evidence synthesis including MA and SR in the future. Most importantly, insights on the nutritional prevention of MD may be uncovered in the process. If the results prove meaningful, the method can be applied to anecdotal evidence in future work. Doing so at this stage, however, would make the results difficult to validate on questions for which formalized study is lacking. By instead applying the proposed approach to the scientific literature on the topic of nutrition for MD, there is a path to qualifying the results *via* a follow-up literature search. Results will also be compared with the current state-of-the-art in topic modeling.

The rest of this article is organized as follows. 'Materials & Methods' will formalize the definition of TF-IDF, the proposed method, key differences between the proposed method and TF-IDF, the data used, pre-processing methods, experimental structure, and the state-of-the-art baseline implemented to contextualize results. 'Results' will detail the results. In 'Discussion', an interpretation of the results is provided. Limitations are also discussed. Finally, 'Conclusions' summarizes all findings and considers future research directions.

## MATERIALS & METHODS

### TF-IDF (background introduction)

TF-IDF is an extensively used statistical approach for topic modeling (including recently). As of 2015, roughly 83% of text-based digital recommender systems are thought to rely on TF-IDF (*Beel et al., 2015*). Therefore, one cannot make the argument that TF-IDF is outdated; it is still used thanks to its efficiency, reliability, and flexibility for topic modeling use cases. TF-IDF is not directly applied in the proposed method, but understanding it is essential for background. The intuition behind TF-IDF is as follows: in a given document, the most topic-significant terms occur frequently in the document while appearing infrequently across the corpus. Note "keywords," "terms," and "topics" can be used interchangeably in this context. Tempering a term's score by its distribution across the corpus accounts for some terms occurring more frequently in general, independently of subject matter. Therefore, a term's uniqueness to a particular document and its internal frequency within that document is a proxy for its topic relevance. Formally, TF-IDF yields a relative frequency score $tf(t, d)$ for a given term t in the document d according to the following classical definition (*Joachims, 1997*):

$$tf(t,d) = f(t,d) * log(|D|/DF(t)) \tag{1}$$

where $f(t, d)$ is the raw number of times the term t appears in document $d$, $|D|$ is the total number of documents in the corpus, and $DF(t)$ is the number of documents in which the term t occurs at least once. The higher the value of $tf(t, d)$, the greater the likelihood of term t representing a significant topic in the document.

### Proposed method (main contribution)

The proposed method in some ways draws from TF-IDF but is not simply concerned with extracting the primary topics from a given document. Instead, the interest is in topics that:

  1. Have the highest proportional occurrence in reports of significant results for the outcome of interest (out of total occurrences of the term across all documents).

  2. Have the most consistent distribution across reports of significant results for the outcome of interest.

  3. Represent "studyable" factors (rather than incidental verbiage).

  Such topics are referred to as potential determinants in this article, as stated in 'Contributions'. As mentioned, the proposed method inherently requires two corpora, one comprising significant results and the other negative or insignificant results on the same general question. These will respectively be referred to as *Cp* and *Cn* going forward.

  Table 1 introduces the proposed method's term scoring structure in terms of the same attributes used in TF-IDF below.

  A notable difference between TF-IDF and the proposed approach is the use of a term's document distribution. In TF-IDF, which assumes only one corpus, there is an inverse correlation between a term's score and its distribution across documents. In the proposed method, however, greater distribution throughout *Cp* instead boosts a given term's score, but not throughout *Cn*.

**Table 1 A comparison between the proposed approach and TF-IDF.** A comparison of TF-IDF and the proposed approach. $Cp$ represents the corpus containing reports of significant and positive findings, and $Cn$ represents the corpus containing reports of insignificant/negative findings.

| Attribute | TF-IDF usage | TF-IDF score correlation | Proposed method usage | Proposed method score correlation |
|---|---|---|---|---|
| Term frequency | Term's frequency in a document | Direct | Proportion of term's occurrences in $Cp$ out of all term occurrences ($Cp$ and $Cn$ inclusive) | Direct |
| Term distribution | Term's distribution across the corpus | Inverse | Term's document distribution throughout $Cp$ only | Direct |

Outside of scoring, candidate topics are filtered for lexical membership in a list. The list contains candidate terms within a category of interest (*e.g.*, nutritional compounds). Regardless of the score, topics must belong to the list to be included in the final results, adding specificity to the search.

Formally, the proposed approach is defined as follows:

Let $DCp$ represent the number of documents in the corpus $Cp$:

$$DCp = |\{D \in Cp\}| \tag{2}$$

where $D$ represents documents.

Let $DCp(t)$ denote the number of documents in the corpus $Cp$ in which a given term $t$ occurs at least once:

$$DCp(t) = |\{D \in Cp : t \in D\}| \tag{3}$$

Let $n(t, Cp)$ represent the raw total count of a given term $t$'s occurrences across documents in Cp only.

Let $n(t, Cp, Cn)$ denote the raw total count of a term $t$'s occurrences across all documents, inclusive of $Cp$ and $Cn$.

Let $b(t)$ represent the proportional occurrence of a given term $t$ specifically in $Cp$ out of all occurrences:

$$b(t) = (n(t, Cp)/n(t, Cp, Cn)) \tag{4}$$

Therefore, the score $a(t)$ for a given term t in $Cp$ is defined as

$$a(t) = \frac{1}{2}(b(t) + (DCp(t)/DCp)) \tag{5}$$

where possible scores range from 0 to 1. A higher $a(t)$ value signals greater topic importance as a potential determinant. Terms must be filtered for membership in a qualifying list of candidates to search for, simply defined as the list . Filtering must also be performed to ensure the $b(t)$ value is above 0.5. This ensures each candidate term already has greater proportional occurrence in $Cp$ with the distribution serving as a secondary criterion.

Note the *proportion* of a term's occurrence in $Cp$ out of all occurrences is used, but its raw count is not (unlike TF-IDF, which merely uses the raw count of a term within a document). This is because potential determinants may not necessarily be the most frequent terms in $Cp$ while still having a unique association with the outcome of interest, particularly if accompanied by considerable distribution throughout $Cp$. Therefore, its

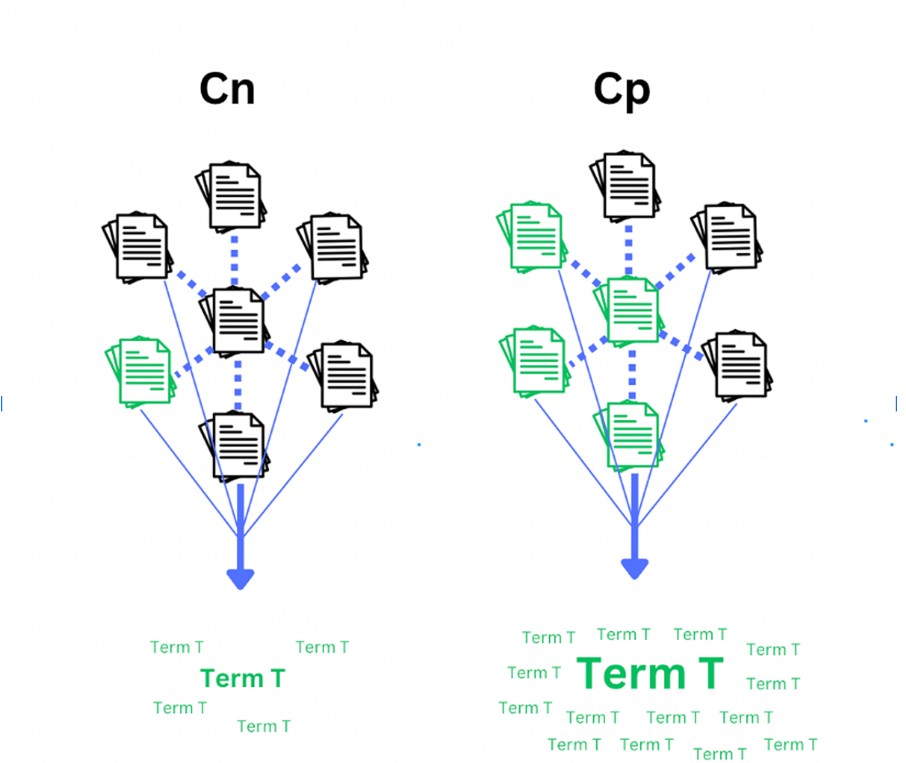

**Figure 1** **Strong distribution and proportional occurrence of a given term in *Cp*.** A graphical representation of a given term T having a strong likelihood of being a potential determinant topic based on the relationship between corpus distribution and term frequency as defined in the proposed method. The green documents represent those containing the term at least once, while the word clouds below the corpora illustrate hypothetical proportional total counts of the term's occurrence in *Cp* vs *Cn*. Note, the example term has both greater distribution and proportional occurrence in *Cp*. This would be considered an ideal scenario and would score highly under the proposed method.

proportional occurrence in *Cp* is more of interest than its raw count. Simultaneously, using proportional values serves as a means of score normalization. Figures 1, 2, and 3 illustrate various scenarios to elucidate the nuances of the proposed method's scoring system.

## Study selections

The following studies were used to test the proposed method. A careful manual selection was made to satisfy three assumptions for studies in *Cp* and *Cn*:

1. Consistent focus on the same objective, namely assessing the impact of nutritional compounds on the development or progression of MD.
2. Broad scope (*e.g.*, the effects of *nutritional supplements, diets,* or *antioxidants* on MD) *vs.* narrow scope (*e.g.*, the effects of *beta-carotene* on MD).
3. Conclusive outcomes clearly categorized as either significant and positive findings or insignificant/negative findings. Figure 4 is a PRISMA flow diagram of the study selection process.

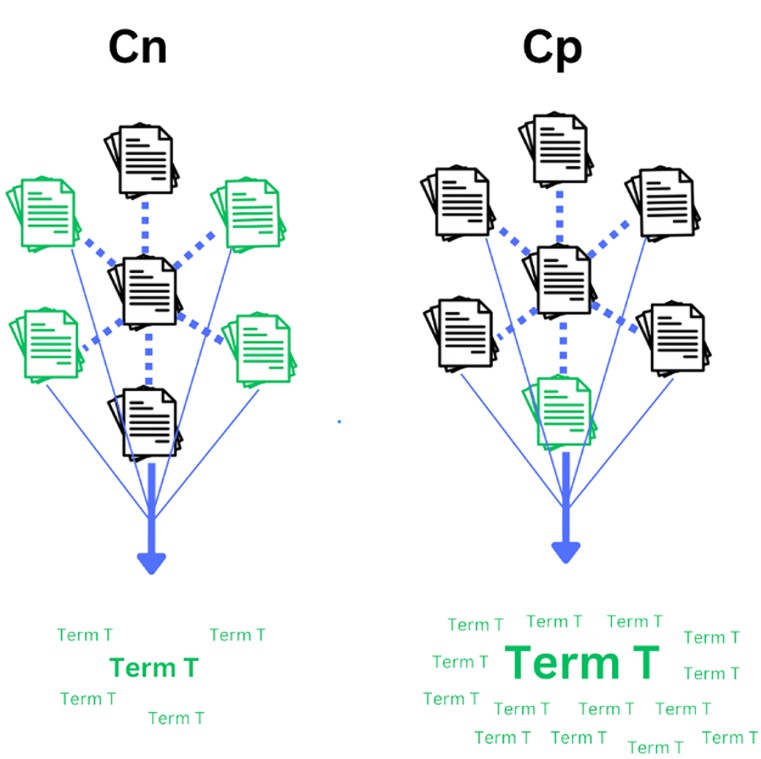

**Figure 2  Weak distribution and strong proportional occurrence of a given term in *Cp*.** A representation of a given term T having a lower probability of being a potential determinant compared to the term in the example from Fig. 1. Note, the total count of term T's occurrences in *Cp* is still higher than that of *Cn*, yet its distribution throughout *Cp* is lower. In this hypothetical scenario, the higher count in Cp is only attributable to one document. The term's score, compared to the Fig. 1. example, would be tempered by the lower distribution in *Cp*.

4. A unique set of entries. Some secondary literature was included. SRs are treated as standalone entries since each comprises a unique set of study selections, interpretations, and conclusions.

To search for relevant articles, the keywords "macular degeneration prevention" and "macular degeneration treatment" were used as base keywords. Each of these base keywords were respectively combined with "nutritional", "nutrition", "nutrient", "nutrients", "antioxidants", "antioxidant", "vitamin, "vitamins", "multivitamin", "dietary", "diet", "food" and "foods. The search only included journal articles.

Note, there were many on-topic studies that either had too narrow a scope (focusing on a single compound for MD) or findings that were too inconclusive to be suitable for comparative analysis. Within the criteria, all apparent qualifying studies were included to avoid selection bias to the degree possible following an extensive literature search (as of August 2023).

Studies in *Cp* contained the following (see Table 2):

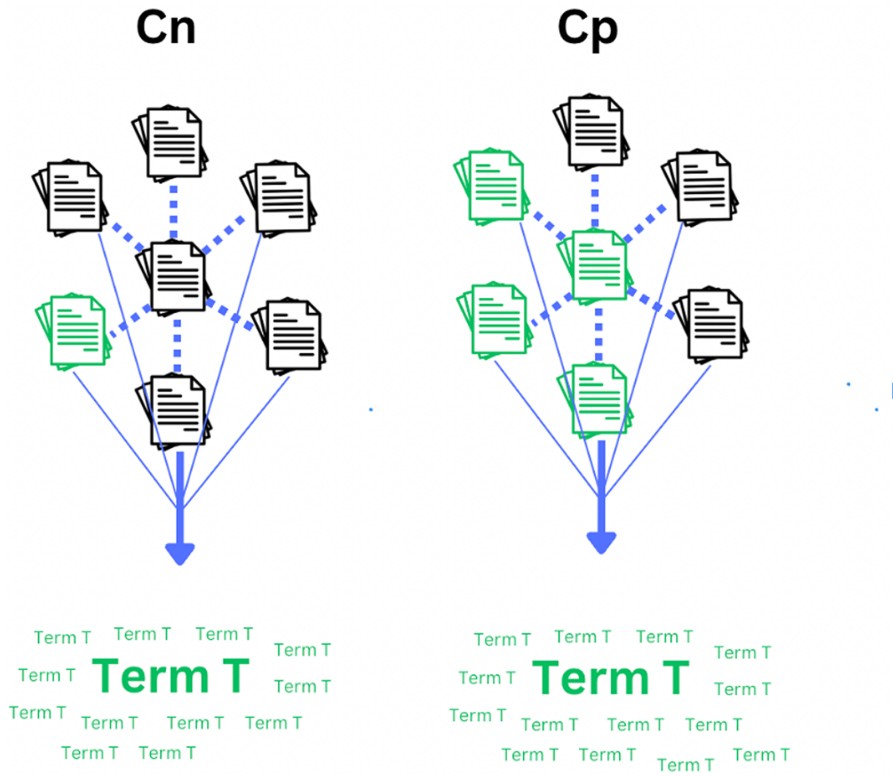

**Figure 3** **Strong distribution and equal proportional occurrence of a given term in *Cp*.** A scenario in which a given term T has roughly equal proportional occurrence in *Cp* and *Cn* in terms of count (only slightly in favor of *Cp*), yet still has a more consistent distribution throughout *Cp*. This term would theoretically score lower than the example in Fig. 1.

Studies in *Cn* comprised the following (see Table 3):

Documents in *Cp* had a combined length of 64,453 words. Documents in *Cn* had a combined length of 65,591 words. Note, the latter refers to total word counts, not the unique set of terms.

## Data pre-processing and testing

Text normalization was performed according to the following steps applied to each document in *Cp* and *Cn*:

1. Lemmatization (*Cambridge University Press, 2009*) was applied to each term in every document across Cp and Cn. This process refers to the reduction of a term to a base form, *e.g.*, "researched" and "researching" to simply "research." Without lemmatization, the various forms of a term would be treated as altogether separate topics, confounding results. Lemmatization was performed *via* Natural Language Toolkit (*NLTK Project, 2023a*), a library for Python.

2. Removal of stopwords *via* Natural Language Toolkit's stopword removal method (*NLTK Project, 2023b*). Stopwords are a set of the most frequently used terms in English (regardless of subject matter), such as "and," "it," or "the."

3. Reduction of all words to lowercase form.

**Table 2  Selection of studies reporting significant results for the prevention or treatment of MD *via* nutritional supplementation.**

| No. | Title | Reference |
|---|---|---|
| 1. | Multivitamin-multimineral supplements and eye disease: age-related macular degeneration and cataract | *Seddon (2007)* |
| 2. | Dietary nutrient intake and progression to late age-related macular degeneration in the age-related eye disease studies 1 and 2 | *Agrón et al. (2021)* |
| 3. | Nutrition effects on ocular diseases in the aging eye | *Chew (2013)* |
| 4. | The impact of supplemental antioxidants on visual function in nonadvanced age-relate macular degeneration: a head-to-head randomized control trial | *Akuffo et al. (2017)* |
| 5. | Adherence to the Mediterranean Diet and progression to late age-related macular degeneration in the age-related eye disease studies 1 and 2 | *Keenan et al. (2020)* |
| 6. | Effect of 2-year nutritional supplementation on progression of age-related macular degeneration | *Piatti et al. (2020)* |
| 7. | Dietary flavonoids and the prevalence and 15-y incidence of age-related macular degeneration | *Gopinath et al. (2018)* |
| 8. | The potential role of dietary antioxidant capacity in preventing age-related macular degeneration | *Arslan, Kadayifçilar & Samur (2019)* |
| 9. | Dietary cartenoids, Vitamins A, C, and E, and advanced age-related macular degeneration | *Seddon (1994)* |

**Table 3  Selection of studies reporting negative or insignificant results for the prevention or treatment of MD *via* nutritional supplementation.**

| No. | Title | Reference |
|---|---|---|
| 1. | Progression from no AMD to intermediate AMD as influenced by antioxidant treatment and genetic risk: an analysis of data from the age-related eye disease study cataract trial | *Awh, Zanke & Kustra (2017)* |
| 2. | Antioxidant vitamin and mineral supplements for preventing age-related macular degeneration | *Evans & Lawrenson (2017)* |
| 3. | Effects of multivitamin supplement on cataract and age-related macular degeneration in a randomized trial of male physicians | *Christen et al. (2014)* |
| 4. | A randomized study of nutritional supplementation in patients with unilateral wet age-related macular degeneration | *García-Layana et al. (2021)* |
| 5. | The value of nutritional supplements in treating age-related macular degeneration: a review of the literature | *Mukhtar & Ambati (2019)* |
| 6. | Dietary antioxidants and primary prevention of age related macular degeneration: systematic review and meta-analysis | *Chong et al. (2007)* |
| 7. | The role of nutritional supplements in the progression of age-related macular degeneration | *van Agtmaal (2014)* |

4.  Punctuation removal.

After applying the above pre-processing steps, each document in *Cp* and *Cn* was treated as a multiset of normalized terms. Figure 5 demonstrates the text normalization process.

The unique set of terms present across all documents in $Cp$ was used for testing. Formally, the set of test terms $B$ is defined as

$$B = \{t \in D : D \in Cp\} \tag{6}$$

where $t$ represents terms and D represents documents.

For each term $t$ in the set $B$, an $a(t)$ score was generated as described in 'Proposed Method (Main Contribution)'.

As specified earlier, topics are filtered for membership within a list $q$. For this experiment, $q$ contained a lexicon of nutritional compounds to search for. This list was obtained from FooDB (*FooDB, 2023*), which offers a database of over 3,000 detected and quantified nutritional compounds. FooDB is supported by the Canadian Institutes for Health Research (*Government of Canada, 2023*). The text normalization steps detailed above were applied to each term obtained from FooDB.

Only the terms with $b(t)$ values above 0.5 were considered. Further, only the terms with final $a(t)$ scores above the 75th percentile were considered for final analysis. Terms with scores in the said range were then filtered for membership in $q$.

Results were contextualized by performing traditional topic modeling on the documents in $Cp$. To apply TF-IDF, the following steps were applied for each term identified by the proposed method:

1. For each document in $Cp$, the said term's TF-IDF score ranking (if present in the document) was calculated using the internal frequency of the term in that document with respect to its document frequency across $Cp$. The ranking is handled such that the most prominent topic has a rank of 1, the second most prominent has a rank of 2, et cetera.

2. The same was done for the same term but this time using each document in $Cn$, while using $Cn$ to establish document frequency.

3. The mean TF-IDF ranking for the term was established both in $Cp$ and $Cn$, respectively.

To further contextualize results with a more recent baseline, Top2Vec (*Angelov, 2020*; *Karas et al., 2022*) was applied to $Cp$ and $Cn$. Top2Vec uses document and word semantic embeddings (*Gutiérrez & Keith, 2019*) to automatically identify the salient topics in a group of documents without specifying the number of topics to expect, as is the case with LDA (*Blei, Ng & Jordan, 2003*). Using a recent model that leverages document and word embeddings is an important step in relating results to the current state-of-the-art. Top2Vec yields groups of terms that are clustered according to their topic association, and has been shown to provide added benefits compared to LDA and TF-IDF. In terms of Top2Vec's results, each resulting cluster represents a salient topic, and the words they comprise are associated within the said topic. Top2Vec was applied separately to $Cp$ and $Cn$ to compare results between the corpora and determine if the proposed approach introduces any added benefit when applied for comparative topic modeling. Top2Vec was implemented using the pre-trained universal-sentence-encoder (*Angelov, 2020*). Note, the Top2Vec implementation in the Python library used does not require stopword removal or lemmatization (*Python Package Index (PyPI), 2024a*). Top2Vec was therefore applied to the raw documents in keeping with the intended use specified in the library documentation.

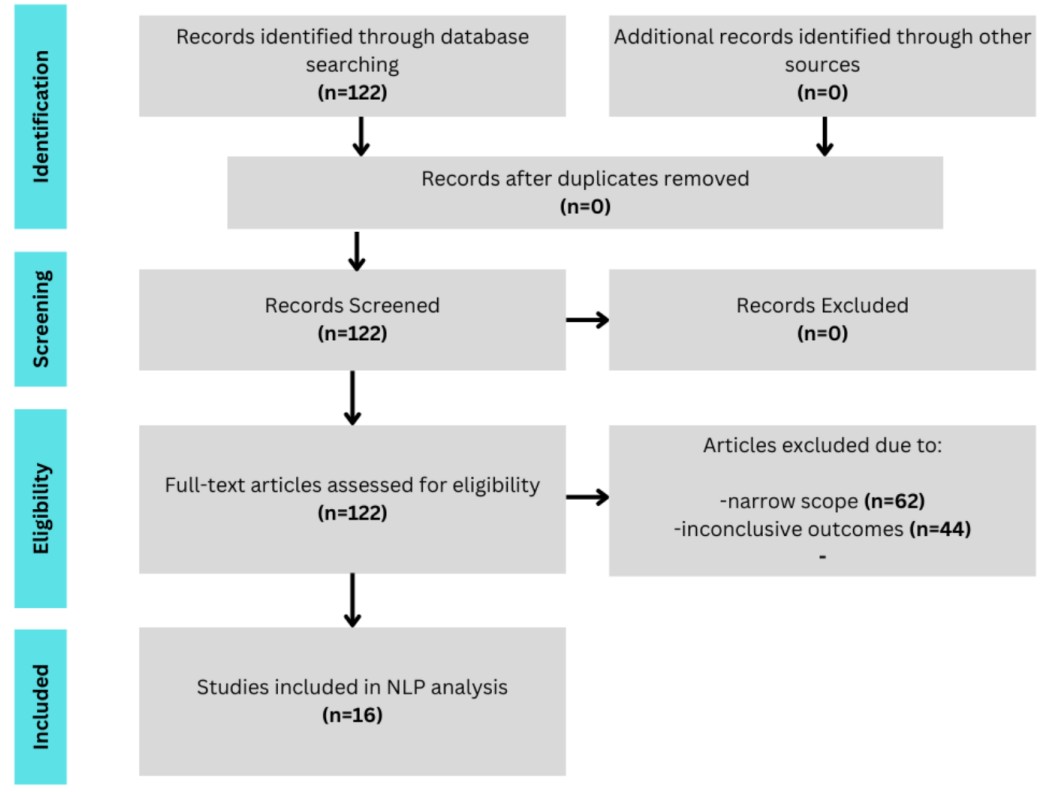

**Figure 4** **A PRISMA flow diagram of the search protocol for relevant studies.**

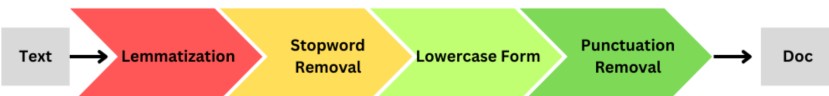

**Figure 5** **Data preprocessing and standardization steps.** An outline of the text standardization steps during data preprocessing.

As another additional baseline, BERTopic (*Grootendorst, 2022*) was applied to each respective corpus as with Top2Vec. BERTopic generates document embeddings *via* transformer-based models, clusters these embeddings, and outputs topic representations *via* a TF-IDF based procedure. BERTopic was implemented using the pre-trained "paraphrase-MiniL3-v2" model (*Python Package Index (PyPI), 2024b*).

## RESULTS

There were 7,119 unique terms in the test set *B*. The distribution of $a(t)$ scores for each of these terms is represented in Figs. 6 and 7 below.

As stated in 'Data Pre-processing and Testing', terms of interest for final analysis are those with $a(t)$ scores above the 75th percentile value (in this case, .55), membership in the list *q*, and $b(t)$ scores above 0.5. Six terms meet these criteria, displayed in Table 4:

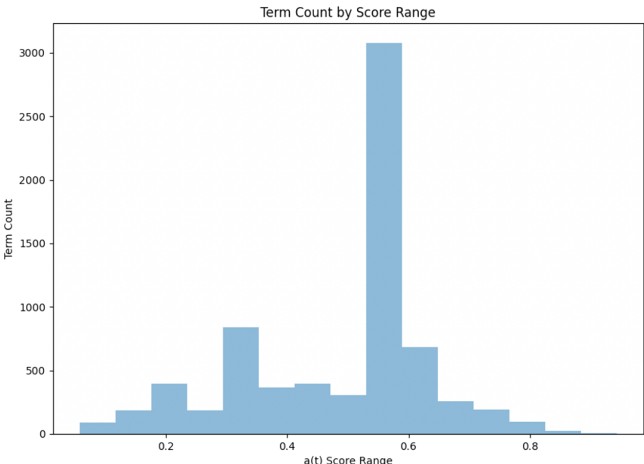

**Figure 6** A histogram of $a(t)$ term score distribution (M = 48, SD = .15).

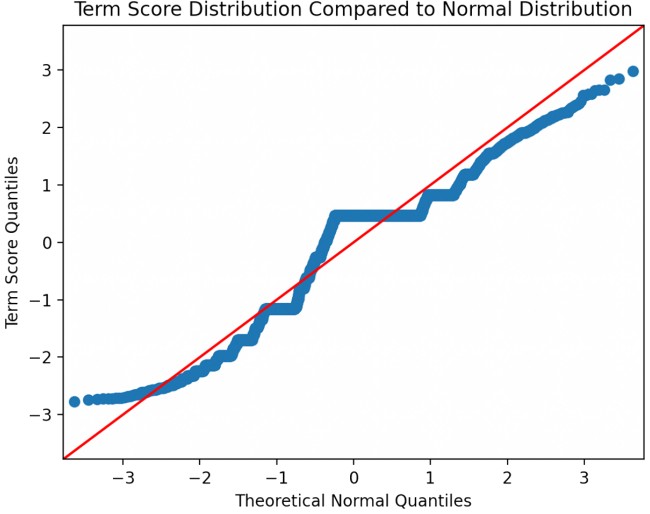

**Figure 7** A Q-Q plot of the term score distribution.

**Table 4** Final selection of compounds found to be especially associated with significant results after filtering for qualifying criteria.

| Term | $a(t)$ Score |
| --- | --- |
| Omega-3 | .78 |
| Copper | .76 |
| Zeaxanthin | .75 |
| Nitrates | .66 |
| Niacin | .66 |
| Molybdenum | .61 |

**Table 5** Mean TF-IDF rankings in Cp documents for each term yielded by the proposed method.

| Term | Mean TF-IDF ranking in *Cp* |
|------|------------------------------|
| Omega-3 | #1584 |
| Copper | #550 |
| Zeaxanthin | #1563 |
| Nitrates | #820 |
| Niacin | #169 |
| Molybdenum | #608 |

**Table 6** Mean TF-IDF rankings in *Cn* documents for each term yielded by the proposed method.

| Term | Mean TF-IDF ranking in *Cn* |
|------|------------------------------|
| Omega-3 | #2076 |
| Copper | #1104 |
| Zeaxanthin | #2126 |
| Nitrates | N/A |
| Niacin | N/A |
| Molybdenum | N/A |

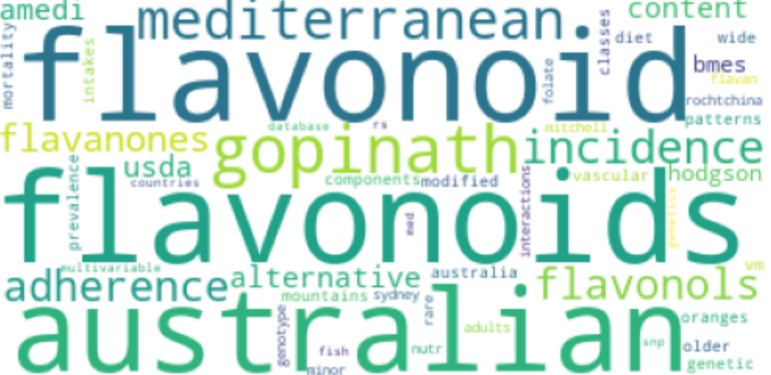

**Figure 8** First topic cluster yielded by Top2Vec when applied to *Cp*.

Table 5 shows the mean TF-IDF rankings for the same terms in *Cp*.

Table 6 shows the mean TF-IDF rankings for the same terms in *Cn*.

Note, nitrates, niacin, and molybdenum show N/A for mean TF-IDF rankings in *Cn* since they are not mentioned in *Cn*.

Figures 8 and 9 show the results of Top2Vec applied to *Cp*. There were two topic clusters identified, each containing an array of associated terms:

Figures 10 and 11 show the results of Top2Vec applied to *Cn*. As with *Cp*, two clusters were identified:

Tables 7 and 8 show the results of BERTopic applied to the respective corpora:

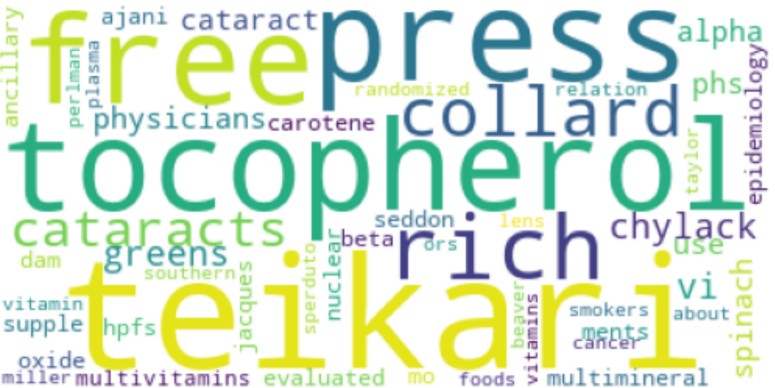

**Figure 9** Second topic cluster yielded by Top2Vec when applied to *Cp*.

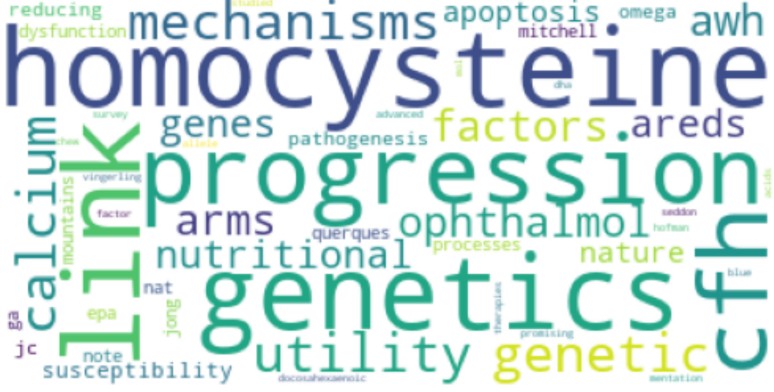

**Figure 10** First topic cluster yielded by Top2Vec when applied to *Cn*.

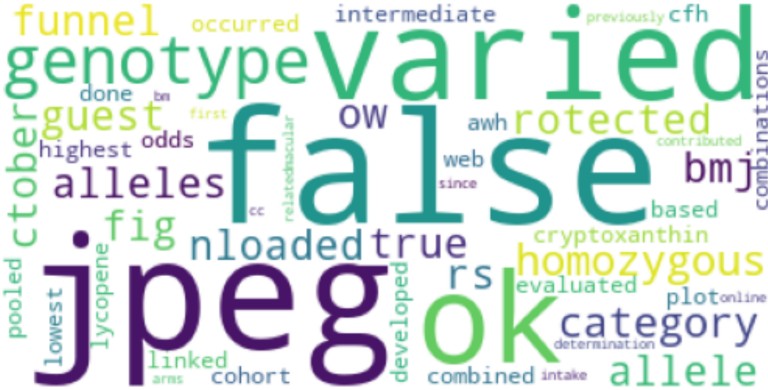

**Figure 11** Second topic cluster yielded by Top2Vec when applied to *Cn*.

**Table 7  BERTopic results on *Cp*.**

| Topic # | BERTopic representation in |
|---|---|
| 1 | '20', 'http', 'com', 'elsevier', '6420', 's0161', 'refhub', 'amd', '30836', 'study' |
| 2 | 'amd', 'age', 'study', 'related', 'group', 'risk', 'intake', 'total', 'dietary', '05' |

**Table 8  BERTopic results on *Cn*.**

| Topic # | BERTopic representation in |
|---|---|
| 1 | 'amd', 'al', 'et', 'age', 'related', 'study', 'risk', 'degeneration', 'macular', 'areds' |
| 2 | 'amd', 'vitamin', 'data', 'study', '10', 'published', 'cochrane', 'risk', 'age', 'related' |

## DISCUSSION

The initial goal of the experiment was to determine whether the proposed method can identify potential determinants (in this case, nutritional compounds) most associated with the outcome of interest, namely the significant prevention or treatment of MD. Six such compounds were identified. Unanimous conclusions across studies are rare due to variations in methodology, experimental structure, participant demographics, or other reasons. Additionally, synergistic effects may make for benefits only when nutritional compounds are administered together (*Khoo et al., 2019*). Therefore, absolute statements cannot be made on individual compounds driving results. However, the follow-up literature search aims to assess the supportability of the findings broadly.

The proposed method identifies omega-3s, copper, zeaxanthin, nitrates, niacin, and molybdenum as more associated with reports of significant outcomes for benefiting MD than other results. *Ma et al. (2012)* find zeaxanthin supplementation to have a preventative effect on the progression of MD. Kelvay (*Cunningham, Cahyadi & Lengyel, 2021*) suggests there is evidence for copper deficiency being associated with MD and that calcium delays MD onset by increasing copper metabolism. Direct supplementation with copper is therefore suggested in the aforementioned study. *Chew et al. (2022)* determine that omega-3 fatty acids added to the AREDS2 (*Chew et al., 2012*) supplement formula come with increased protective effects, especially for late-stage MD. *Jiang et al. (2021)* find high omega-3 fatty acid intake is associated with a 14% reduction in risk of developing early MD and a 29% reduction for late MD. *Mrowicka et al. (2022)* explore the mechanisms by which zeaxanthin supplementation can prevent MD. *Broadhead et al. (2023)* show that higher nitrate intake (including from leafy green vegetables) is associated with a lower risk of progression to late MD. Building on the latter finding, *Larsen (2023)* suggests adding a nitrate supplement to the AREDS2 supplement formula as a worthwhile research avenue. *Erie et al. (2009)* highlight how lower copper levels in the retinal pigment epithelium are associated with MD and support supplementation for at-risk population members. The follow-up literature search did not find results to support the use of niacin or molybdenum

for MD. Meanwhile, niacin has been shown to have potentially adverse ocular effects (*Fraunfelder, Fraunfelder & Illingworth, 1995*; *Tittler et al., 2008*) while still offering some promise for glaucoma prevention (*Taechameekietichai et al., 2021*).

The proposed method successfully filters thousands of candidate keywords from broad-scope studies down to six nutritional compounds. Four of these are supported by narrow-scope studies in terms of effectiveness based on a follow-up literature search. In this sense, the proposed method delivered as intended by pinpointing specific, supportable potential determinants from a wide range of possible topics within broad-scope studies. This work does not support the notion that the identified compounds are the only effective ones. It only highlights their particular association with positive results, and these effects may be in tandem with other compounds that are typically used to treat/prevent MD including those in the AREDS formulation. Interestingly, the two identified compounds for which the follow-up literature search yielded little to no evidence of effectiveness for MD are also within the lowest range of $a(t)$ scores. Since the term "nitrates" shares a score with "niacin" (0.66), 0.66 may be near the theoretical lower bound of significance with terms in this range yielding mixed results. Nevertheless, this observation loosely substantiates the notion that the magnitude of the $a(t)$ score may be a proxy for the likelihood of a topic representing a potential determinant, with significance decreasing as the score decreases.

When TF-IDF mean rankings are established for the results, none of the terms identified by the proposed method have a particularly prominent ranking (ranging from an average of 169th place to 1,584th place) in $Cp$. However, all of the identified terms had even less prominent TF-IDF rankings across documents in $Cn$ (or were not present at all), supporting the proposed method's results. That said, simply using TF-IDF as described in 'Data Pre-processing and Testing' (namely by taking a each term in a test set and calculating its prominence in every document in $Cp$ and then $Cn$ to get an average ranking for each respective corpus) may be viable but is in theory less efficient than the proposed approach, which instead only needs to make one calculation for every term.

The Top2Vec baseline implementation identified two main topic clusters for $Cp$. The first included terms like "flavonoids" and "mediterranean", presumably referring to the high flavonoid mediterranean diet and associated benefits. The other cluster related tocopherol and various foods like spinach and collard greens, potentially aligning with the work of Broadhead et al. (*Cunningham, Cahyadi & Lengyel, 2021*). For $Cn$, two clusters were similarly returned. The first grouped together terms that are likely mentioned together in the context of risk factor and stressor mechanisms pertaining to MD, including genetics and levels of key amino acids like homocysteine. The second cluster did not have as much of a discernable pattern. The Top2Vec results loosely suggest that studies in $Cp$ focused more on whole-food diets than studies in $Cn$. However, although broad categories like flavonoids were captured, the only specific nutritional compounds that appeared in the Top2Vec topic clusters for $Cp$ were tocopherol and carotene. The proposed method therefore identified more compounds that have particular associations with studies in $Cp$ that may be missed by other models. Such compounds may have a higher distribution and proportional occurrence in $Cp$, but perhaps not enough to always show as a prominent topic with existing topic models that aren't specifically designed to identify potential

determinants. However, no claim is yet made about how results compare to Top2Vec (or other embedding-based models) on larger sample sizes within the scope of this article. As implemented, Top2Vec will pick up information that isn't directly relevant (like the names of cited authors). These results further support the advantages of the proposed method in terms of its ability to inform specific recommendations in the context of comparative topic modeling and the search for potential determinants. The results, however, do not point to the proposed method outperforming Top2Vec in all topic modeling use cases, an important distinction.

BERTopic's results did not yield any specific compounds yet identified general topics like "vitamins" and "areds" (referring to the supplement formula). The sample size may be too small to warrant the application of a transformer-based model. Nevertheless, these results support the proposed method's advantage in another sense, namely, to yield meaningful results on smaller sample sizes.

Currently, limitations of the proposed method include searching for a predefined set of items defined in the list $q$ rather than being category-agnostic. The proposed method will not identify the more nuanced differences in study design but only works within a keyword-based topic modeling context. In its current form, dosages are not treated as topic candidates without significantly modifying the method. It is also possible that a potential determinant is associated with the positive corpus without any causal relationship to the outcome, for instance, by frequently being ruled out as a cause (and mentioned often as such) or discussed without being tested. Scores can also be distorted by the unlikely scenario in which a given term has a high count in $Cn$ compared to $Cp$ (perhaps due to unusually frequent mentions in a single document), yet the remaining instances of the term in $Cp$ still have high document distribution. In the latter hypothetical case, a topic's score would be skewed to the lower side due to a single document in $Cn$ lowering a term's proportional occurrence in $Cp$. Case-by-case interpretation would be required if this were to occur. Another important limitation is study selection. For this experiment, all the qualifying studies (as per criteria in 'Study Selections') that could be found in the literature at the time of conducting this test were included. However, incomplete representation of the existing studies or biased, imbalanced selections can skew results. Finally, additional normalization around counts of documents containing a given term may be needed if there are drastic differences in the total combined length of documents in $Cp$ vs $Cn$.

Fundamental gains introduced by the proposed method include sensitivity to disparity in a topic's importance between two corpora, which underscores potential determinants (drivers of divergence in results). A topic can have great proportional frequency and distribution in the positive corpus compared to the other corpus, yet may still be a minor topic overall such that traditional topic models don't rank it highly (or even at all). Conventional topic models are designed to accurately represent the main topics in documents and corpora. However, just because a topic has a distinct association with reports of significant results (and should be studied further as a driver of said significant results) does not mean it is necessarily among the most prominent topics in the corpora. By instead attuning the proposed model to topics that most distinctly characterize reports of significant results (not the same as finding the most prominent topics), specific insight

is generated. The disparity in frequency of occurrence and distribution may signal a topic's role as a potential determinant, yet it may simultaneously be missed by traditional models including TF-IDF, Top2Vec, and BERT for the reasons explained above. Additional benefits include more meaningful performance on small sample sizes (as suggested by analyzing BERT's results) and better efficiency than applying a model like TF-IDF to every single document and comparing topic rankings.

The main recommendation for practitioners supported by this article's results is the inclusion of omega-3 fatty acids, copper, nitrates, and zeaxanthin in nutritional supplementation intended to prevent or treat macular degeneration. More broadly, these results further demonstrate the value of NLP for MA and SR and point to comparative topic modeling as a relevant and fruitful avenue for evidence synthesis. Further application across domains is encouraged to leverage comparative topic modeling as a means of generating insight. Future studies can apply the method detailed in this article to guide or augment the application of other current models, including LLMs. Anecdotal evidence should also be analyzed with the proposed method to find patterns that are robust enough to warrant further, formalized study, and possibly enable new insights and discoveries. For the baseline model implementations, minimum allowable array sizes may change with new Python library versions, in which case it's important to normalize array sizes across methods at the practitioner's discretion.

In 'Relevant Literature', a review of relevant work showed that the most common applications of NLP for SR and MA pertain to fact/figure mining and literature search. While this work does not constitute SR or MA in the strictest sense, it helps expand the applications of NLP to directly augment the comparative contextualization of evidence (with potential for evidence synthesis like SR and MA) and the substantiation of future research directions on emerging questions.

## CONCLUSIONS

This study explored a method for discovering key topics that may represent determining factors responsible for divergent study results. Under the proposed method, six nutritional compounds are found to have a particular association with studies reporting significant results for preventing or treating MD through dietary supplementation. Upon conducting a follow-up literature search on these compounds, four are shown to be effective in benefiting MD based on a considerable body of studies. These results support the inclusion of omega-3 fatty acids, copper, zeaxanthin, and nitrates in nutritional supplementation for amelioration of MD. The two remaining compounds not supported for effectiveness by the follow-up literature search (niacin and molybdenum) are simultaneously those in the lowest range of scores under the proposed method, substantiating the proposed method's score for a given topic as a practical measure of its interest as a potential determinant. Further, the proposed method identified compounds that were not captured as prominent topics by a state-of-the-art topic model that uses word and document embeddings (Top2Vec). Future direction for this work includes applying the proposed method to emerging research questions to validate new directions of inquiry in a targeted

way, including with larger sample sizes. The proposed method will also be applied to patient experiences and anecdotal reports on diseases that are not well understood, the interest being to identify practices/treatments that are associated with benefits and warrant further, formalized study.

## ACKNOWLEDGEMENTS

The author extends thanks to his family for their ongoing support.

### Funding
The authors received no funding for this work.

### Competing Interests
The authors declare there are no competing interests.

### Author Contributions
- Lucas Jacaruso conceived and designed the experiments, performed the experiments, analyzed the data, performed the computation work, prepared figures and/or tables, authored or reviewed drafts of the article, and approved the final draft.

### Data Availability
The Python code used is available in the Supplemental File.

The FooDB dataset (which contains a list of known nutritional compounds used in the process detailed in the article) is available at: https://foodb.ca/compounds.

### Supplemental Information
Supplemental information for this article can be found online at http://dx.doi.org/10.7717/peerj-cs.1940#supplemental-information.

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
