# Peer review of "Insights into the nutritional prevention of macular degeneration based on a comparative topic modeling approach"

_PeerJ Computer Science, doi:10.7717/peerj-cs.1940_

## Round 0.1 · original submission · Major Revisions

The authors underline that the manuscript has merit although it should be revised in terms of experimental design for explaining some choices made in the manuscript.

Reviewer 1 ·

Basic reporting

The paper under review presents the formulation of a novel comparative topic modeling approach to analyze reports of contradictory results on a general research question. The authors investigate various techniques and in Macular Degeneration prevention topic.
The research is well-motivated and addresses an important problem in the field. Overall, the manuscript is well written, but the clarity can be improved adding a table with the list of abbreviations and using higher quality figures (8-11).

Experimental design

More various and meaningful baselines could be used such as transformers based methods.

Validity of the findings

I think there is a problem with lemmatization: in figure 8 there are "flavonoid" and "flavonoids".

Reviewer 2 ·

Basic reporting

The purpose of this research, as outlined in the article, is to give a methodology for identifying crucial aspects that may influence the varying outcomes of different studies. Word embeddings are a method for numerically representing words and entire phrases.
Word2Vec is a widely used paradigm for acquiring distributed representations of words that effectively capture both their semantic and syntactic meanings. Doc2Vec is an extension of the Word2Vec paradigm that is designed to learn representations at the document level. Top2Vec is a model that is designed to acquire dispersed representations of subjects within a collection of documents.
Top2Vec is a computational method that identifies and generates vectors for subjects, documents, and words, all embedded together. This study presents a comparative topic modeling method as an alternative to Top2Vec. The purpose is to assess reports that have inconsistent outcomes on the same research question.
The authors of the study attempted to find themes that showed varying connections with significant impacts for a certain outcome. They ranked these topics based on their proportional incidence in reports of significant effects and the consistency of their distribution.
The work is undeniably intriguing since it presents an alternative approach to word representation methodologies. The paper also includes significant methodological details.
However, the article in its current form is not acceptable for publication due to significant presentation issues, lack of methodological explanations, and lack of a more detailed discussion. Below, I provide my comprehensive remarks.
• The anecdotes defended between lines 100 and 114 under the Introduction section should be supported with appropriate references.
• I think the literature review given under the Introduction section is sufficient. Only one or two examples of the widespread use of LDA in MA and SR research can be given. Because LDA appears as the most remarkable model in MA and SR research.
• The anecdotes presented between lines 195 and 223 under the Contributions subheading under the Introduction section should be supported with appropriate references.
• The division of the study is well done, and the English grammar and expression of the article content are at a satisfactory level.

Experimental design

• Selecting the articles depends on which search string you used and from which bibliometric data source you obtained these articles?
• How did you apply criteria such as article type, journal article, conference paper, book chapter?
• What fundamental contributions and gains does the proposed approach provide compared to Top2Vec and other word embedding methods such as Word2Vec, TF-IDF, GloVe, FastText, ELMO, CoVe, BERT, RoBERTa? I suggest you give a two or three sentence explanation.
• I did not see the experimental data set as a text collection in the additional files, you need to add it.

Validity of the findings

• Figures 8 and 9 also show the plural forms of the words. For example, “Flavonoid” and “Flavonoids” in Figure 8. If the lemmatization process is applied to phrases more accurately, words from the same root can be represented with a single word and it will reduce the size of the word vector.
• Combining Figure 1, Figure 2 and Figure 3 and highlighting their differences and presenting them in a single figure can save the readers from confusion.
• The discussion is the most superficial element of the article, and I anticipate that this section will be greatly deepened before it is published. The commentary at this time consists solely of a re-listing of the results that are presented in the tables. There is no more explanation provided regarding the possible suggestions for researchers and practitioners, nor is there any reason provided for the results that were discovered.
• The authors could enhance the contextualization of the study's impact and develop more practical guidance for researchers and practitioners, to ensure its applicability in real-world settings.
• The limitations of the study should be clearly stated and guidelines for future studies on this subject should be suggested.

---

## Round 0.2 · accepted · Accept

The reviewers acknowledge that the manuscript has been improved according to their suggestions.

Reviewer 1 ·

Basic reporting

Overall, the manuscript is well-written and provides valuable insights into the current state of research. However, there are a few areas where further clarification or elaboration could enhance the manuscript's impact and readability. Additionally, discussing alternative interpretations of the findings would contribute to a more comprehensive evaluation of the research.

Experimental design

The experimentation has been enhanced with the introduction of new baselines.

Validity of the findings

The author has generously included the code; however, it is presented in a non-optimal format as a .txt file. A more effective solution could have been the provision of code in notebook format. Notebooks offer several advantages, including enhanced readability, interactivity, and the ability to include detailed explanations alongside the code. Therefore, transitioning the code to notebook format would greatly enhance its utility and impact.

Reviewer 2 ·

Basic reporting

I think that the authors adequately addressed the corrections I mentioned and thus resolved my concerns.

Experimental design

I think that the authors adequately addressed the corrections I mentioned and thus resolved my concerns.

Validity of the findings

I think that the authors adequately addressed the corrections I mentioned and thus resolved my concerns.

Additional comments

I think that the authors adequately addressed the corrections I mentioned and thus resolved my concerns. Moreover, the method and results supported by the BERT topic added even more richness of meaning to the article. I believe it would be appropriate to publish the article in this updated form.